# Cytokine Imbalance as a Biomarker of Treatment-Resistant Schizophrenia

**DOI:** 10.3390/ijms231911324

**Published:** 2022-09-26

**Authors:** Natalia A. Shnayder, Aiperi K. Khasanova, Anna I. Strelnik, Mustafa Al-Zamil, Andrey P. Otmakhov, Nikolay G. Neznanov, German A. Shipulin, Marina M. Petrova, Natalia P. Garganeeva, Regina F. Nasyrova

**Affiliations:** 1Institute of Personalized Psychiatry and Neurology, Shared Core Facilities, V.M. Bekhterev National Medical Research Centre for Psychiatry and Neurology, 192019 Saint Petersburg, Russia; 2Shared Core Facilities “Molecular and Cell Technologies”, V.F. Voino-Yasenetsky Krasnoyarsk State Medical University, 660022 Krasnoyarsk, Russia; 3International Centre for Education and Research in Neuropsychiatry, Samara State Medical University, 443016 Samara, Russia; 4Department of Psychiatry, Narcology and Psychotherapy, Samara State Medical University, 443016 Samara, Russia; 5Department of Physiotherapy, Faculty of Continuing Medical Education, Peoples’ Friendship University of Russia, 117198 Moscow, Russia; 6Basic Department of Psychological and Social Support, St. Petersburg State Institute of Psychology and Social Work, 199178 Saint Petersburg, Russia; 7St. Nikolay Psychiatric Hospital, 190121 Saint Petersburg, Russia; 8Centre for Strategic Planning and Management of Biomedical Health Risks Management, 119121 Moscow, Russia; 9Department of General Medical Practice and Outpatient Therapy, Siberian State Medical University, 634050 Tomsk, Russia

**Keywords:** cytokines, cytokine status, treatment-resistant schizophrenia, biomarker, chronic neuroinflammation

## Abstract

Treatment-resistant schizophrenia (TRS) is an important and unresolved problem in biological and clinical psychiatry. Approximately 30% of cases of schizophrenia (Sch) are TRS, which may be due to the fact that some patients with TRS may suffer from pathogenetically “non-dopamine” Sch, in the development of which neuroinflammation is supposed to play an important role. The purpose of this narrative review is an attempt to summarize the data characterizing the patterns of production of pro-inflammatory and anti-inflammatory cytokines during the development of therapeutic resistance to APs and their pathogenetic and prognostic significance of cytokine imbalance as TRS biomarkers. This narrative review demonstrates that the problem of evaluating the contribution of pro-inflammatory and anti-inflammatory cytokines to maintaining or changing the cytokine balance can become a new key in unlocking the mystery of “non-dopamine” Sch and developing new therapeutic strategies for the treatment of TRS and psychosis in the setting of acute and chronic neuroinflammation. In addition, the inconsistency of the results of previous studies on the role of pro-inflammatory and anti-inflammatory cytokines indicates that the TRS biomarker, most likely, is not the serum level of one or more cytokines, but the cytokine balance. We have confirmed the hypothesis that cytokine imbalance is one of the most important TRS biomarkers. This hypothesis is partially supported by the variable response to immunomodulators in patients with TRS, which were prescribed without taking into account the cytokine balance of the relation between serum levels of the most important pro-inflammatory and anti-inflammatory cytokines for TRS.

## 1. Introduction

Schizophrenia (Sch) is a common socially significant mental disorder associated with premature mortality and reduced life expectancy of patients [1,2]. Epidemiological studies show that the life expectancy of patients with serious mental disorders, including Sch, is reduced by 7–24 years [3]. This is partly due to the development of serious adverse reactions to drugs (primarily metabolic syndrome caused by antipsychotics (APs)) and therapeutic resistance to APs [4].

Therapeutic resistance (TR) is a condition in which a mental disorder cannot be treated or corrected despite an adequate course of treatment. Currently, the TR problem remains relevant in relation to a wide range of mental disorders: depressive disorder, obsessive-compulsive disorder, bipolar affective disorder, Sch, etc. [5].

Due to the fact that APs of the first and new generations affect different mechanisms of action in the treatment of Sch, the risk of developing TR to some APs of new generations remains high, almost similar to APs of the first generation. This may be due to the fact that the pathophysiological mechanisms of TRS development are more complex than previously thought. All currently available APs are able to antagonize dopamine D2 receptors, and the APs’ therapeutic effects in psychosis are related to their action on the limbic system reducing dopamine transmission [6]. This is confirmed by several reports that therapeutic doses of typical APs block D2 receptors in 70–89% of cases in young adults, while atypical APs block D2 receptors in 38–63% of cases [7]. In addition to the effect of AP on dopaminergic neurons, other possible mechanisms are being considered [6,8].

TR to APs in patients with Sch or treatment-resistant Sch (TRS) is one of the urgent problems of psychiatry and clinical pharmacology and is far from being resolved, despite the development and use in real clinical practice of new generations of APs (Figure 1). The average incidence of TRS occurs in approximately 30% of individuals diagnosed with Sch [9]. However, according to a meta-analysis by Suzuki et al. [10], the frequency of occurrence of TRS varies from 0 to 76%. This may be due to both the difference in the methodology and design of epidemiological studies of TRS, and different definitions of TRS, which have been revised several times. In 2004, the American Psychiatric Association considered TRS to be a minor or no symptomatic response to multiple (at least two) APs given for an adequate duration (at least 6 weeks) and in an adequate therapeutic dose (therapeutic range), while at least one AP must be of a new generation [11]. The 2012 World Federation of Societies for Biological Psychiatry (WFSBP) Biological Treatment of Sch guidelines define TRS as a disorder in which no significant improvement in psychopathological symptoms and/or other target symptoms has been achieved despite treatment with at least two different APs from two different chemical classes (at least one must be an atypical AP) at recommended therapeutic doses for a treatment period of at least 2–8 weeks [12]. The National Institute for Health and Clinical Excellence (NICE) in 2014 defined the criteria for TRS as insufficient response to at least two different consecutively prescribed APs at appropriate doses taken over an appropriate period of time (4–6 weeks); however, at least one AP must be new-generation non-clozapine APs [13]. According to the Diagnostic and Terminology Working Group Guidelines, Treatment Response and Resistance in Psychosis (TRRIP), TRS is considered to be at least moderately severe, with <20% improvement in Sch symptoms, at least moderate functional impairment (based on the appropriate approved scale), and confirmed adherence to APs by measuring the concentration of APs by taking two different APs at adequate therapeutic doses for at least 6 weeks, while at least one AP is a prolonged injectable form [14,15]. In addition to the above criteria, TRS criteria have also been proposed by other Sch treatment guidelines, such as The Texas Treatment Algorithm Project [16] and the International Psychopharmacological Algorithm Project (IPAP, 2006) [17]. All these definitions of TRS are different and subject to a wide range of interpretations, which can lead to inconsistent clinical management and inaccurate treatment [18], as well as variable results of epidemiological studies.

Sch and TRS Clinical Experts in 2017 reviewed the main areas of TRS research. They concluded that the diagnosis of TRS required an inadequate response to two different APs, each taken at an adequate dose and for an adequate duration. In each course of treatment, it is recommended to use objective Sch symptom scores to assess response to APs to ensure adherence to APs therapy. Once no response has been established (after ≥12 weeks for positive symptoms [2 courses of ARs lasting ≥6 weeks]), it is recommended that the Sch treatment plan be reviewed and alternative pharmacological or non-pharmacological treatments be considered [9].

In addition, TRRIP has developed current criteria for defining TRS and a consensus has been reached on “minimum requirements”. However, any definition of TRS must indicate that the Sch patient received an adequate course of APs in terms of dosage (equivalent to or greater than 600 mg chlorpromazine per day), two courses of two different APs lasting 6 weeks (each course) at a therapeutic dose, active control of adherence to treatment (≥80% of prescribed doses), as well as the use of structured clinical assessments to establish the presence and severity of symptoms [20].

The mechanisms of TRS development are heterogeneous (Table 1) and have been actively studied for many years, but there is no single view. Several hypotheses have been proposed (Figure 2), among which the TRS inflammatory hypothesis is of particular scientific and clinical interest, since the use of APs is ineffective in approximately 30% of all patients with Sch. This may be due to the fact that some patients with TRS may suffer from pathogenetically “non-dopamine” Sch [21]. In each case of TRS, it is important to rule out pseudo-resistance, the underlying mechanism of which may lie in an imbalance of pro-inflammatory and anti-inflammatory cytokines (cytokine status).

The purpose of this narrative review is an attempt to summarize the data characterizing the patterns of production of pro-inflammatory and anti-inflammatory cytokines during the development of therapeutic resistance to APs and their pathogenetic and prognostic significance of cytokine imbalance as TRS biomarkers.

## 2. Pathogenetic Aspect of Inflammation in Treatment Resistance Schizophrenia

There are many factors leading to a chronic neuroinflammatory process in Sch. Research in this area has led to the formation of several mechanisms of TRS (Figure 3), which will be discussed in detail later.

### 2.1. Changes in the Functional Activity of Microglia in Treatment Resistance Schizophrenia

Microglia account for 10–20% of all cells found in the brain and are an important component of the CNS immune system [47]. Microglia play an important role in neuroinflammation, providing protection in the event of damage or disease to the CNS. When neuroinfection occurs, activation of microglia, synthesis and release of central pro-inflammatory cytokines, which leads to various mental and behavioral disorders [48]. There is now evidence that aging [49], neurodegeneration [50] and stress [51] lead to “sensitization” or “priming” of microglia, which subsequently causes an exaggerated immune response. Exposure of primed microglia to, for example, minor systemic inflammation leads to proliferation and increased production of pro-inflammatory cytokines [52], which, in turn, can exacerbate the immune response in the CNS and be expressed in a change in behavior [53]. One drug thought to reduce microglial activation is minocycline, a broad-spectrum tetracycline antibiotic with broad anti-inflammatory activity [54]. First and second generations of APs regulate the secretory profile of microglia in vitro. They inhibit the release of proinflammatory cytokines from activated microglia and alleviate oxidative stress [55]. However, some recent reports have shown conflicting results on the effect of some APs on the release of pro-inflammatory cytokines [56]. At the same time, not all APs have an anti-inflammatory effect, which may be due to the role of microglia in the development of TRS. 

### 2.2. Sensitization or Kindling in Treatment Resistance Schizophrenia 

“Firing”/“sensitization” refers to the process by which the initial immune response to some stimulus (stress or infection) raises or lowers the threshold to respond to the next exposure to the same stimulus. At the same time, a weaker stimulus is required to activate the immune response or release cytokines than with the initial exposure to an unfavorable (damaging) factor. It is believed that the memory function of the acquired immune system is responsible for this process [51]. The action of factors such as systemic inflammation or stress on healthy people leads to the stimulation of the immune response. As a result, cell proliferation is activated, an increase in the production and release of pro-inflammatory cytokines is observed [57]. “Firing up”/“sensitization” supports the hypothesis that neuroinfection in early childhood may lead to increased release of cytokines when the immune system is activated later in life. These processes lead to neurotransmitter disorders. Stress induces a pro-inflammatory immune response in CNS. However, this is usually reduced after a stressful event. Psychopathological symptoms and neuroinflammation are associated with the immune response of CNS cells to stress, and neuroinflammation is involved in stress-related behavioral changes induced by cytokines and mediated by neurotransmitters. Studies have found that after exposure to chronic stress or repeated stressful events, the threshold for physiological responses of the CNS to stress decreases. As a result, less stimulus is enough to activate an immune or neurotransmitter response. In an animal study, it was shown that with age, the brain is in a sensitized state and produces more cytokines for inflammatory stimuli than the brain of young animals [52]. Repeated exposure to pro-inflammatory cytokines leads to increased neurotransmitter responses [58] as, for example, with tumor necrosis factor alpha (TNF-α) [59]. Stress causes activation and proliferation of microglia in the CNS, which may possibly mediate these cytokine effects [52]. Chronic stress is known to affect the glutamatergic system, ionotropic and metabotropic glutamate receptors and excitatory amino acid transporters [60], which may also play a role in the development of TRS, as it is associated with higher levels of glutamate in the anterior cingulate cortex [61].

### 2.3. Vulnerability-Stress-Inflammation in Treatment Resistance Schizophrenia

The risk of developing TRS increases with stressful life events or psychological stress, especially those that act at key periods in the development of the CNS. The Sch vulnerability-stress model was first proposed by Zubin and Vesna [62], who suggested that stress above the vulnerability threshold in humans contributes to the development of a psychotic episode. It is important to add inflammation to this model, forming the vulnerability-stress-inflammation model, since neuroinflammation plays an important role in the pathogenesis of TRS and can in turn be caused by stress [63]. For example, if an inflammatory response in the CNS is stimulated in a second trimester of pregnancy or offspring while the CNS is still developing, the offspring may be a risk of developing Sch. Animal studies have shown that exposure to stress at an early age leads to an increase in the level of pro-inflammatory cytokines [64], which play an important role in the development of Sch and TRS. Vulnerability-stress-inflammation-induced immune dysregulation is associated with dysregulation of many neurotransmitter systems that APs cannot therapeutically address. Thus, the development of TRS is likely associated with stress-induced inflammation [5].

### 2.4. Prenatal, Perinatal and Postnatal Infection in Treatment Resistance Schizophrenia

Existing epidemiological studies give us the idea that prenatal exposure to maternal infection is associated with an increased risk of Sch in the offspring [65]. The risk of developing Sch may be related to the direct effects of neuroinfection (e.g., disruption of structure due to cyst formation, exposure to inflammatory factors) as well as neurochemical changes such as increased dopamine levels associated with poor performance. Catechol-O-methyl transferase (COMT) and increased dopamine synthesis caused by *Toxoplasma gondii* infection [66]. Exposure to viruses and other infectious agents—influenza, herpes simplex virus type 2, Coronavirus Disease 2019 (COVID-19) during pregnancy and at the time of conception—is associated with a greater risk of psychotic disorders [67]. Given the changes in pro-inflammatory cytokine production in pregnant women with COVID-19, schizophrenic and psychotic disorders may potentially be a long-term risk in the offspring of pregnant women who have experienced COVID-19 [68]. Animal studies have also provided evidence for the role of pre- and perinatal infections in the later development of Sch [69]. For example, after prenatal exposure to viruses, offspring show typical symptoms of Sch, such as cognitive impairment or startle reflex abnormalities [70]. Maternal bacterial infection during pregnancy is closely associated with the development of psychosis in the offspring and varied depending on the severity of the infection and the sex of the offspring. At the same time, the effect of a multisystem bacterial infection was almost two times higher than that of a less severe localized bacterial infection [71]. Of interest are studies that have demonstrated the association of Sch development with prenatal or early childhood exposure to various viruses [72], respiratory infections [73] and infections of the genital organs or reproductive tract [74]. Because Sch develops more frequently during adolescence or adulthood, it is important to establish a possible mechanism for the association between early infection and Sch in adults. Studies in animal models show that early infection or immune activation affects several processes of neurogenesis, including dopaminergic and glutamatergic neurotransmission [75]. The study of bacterial [71] and some other infections in humans [76] are examples that highlight this connection. The risk of developing TRS is also indicated by an increased level of C-reactive protein (CRP) or cytokines in childhood [77]. In addition, neuroinfection at a later age has been shown to be associated with an increased risk of developing TRS. A large epidemiological study conducted in Denmark showed that autoimmune disorders, as well as severe infections, increase the risk of developing Sch and Sch spectrum disorders. This is especially true for patients with both risk factors for TRS [78].

### 2.5. Cytokine Imbalance in Treatment Resistance Schizophrenia

Based on the meta-analyses by Momtazmanesh et al. [79], it is possible to conditionally classify cytokines according to their serum levels in patients with TRS into four groups: group 1—elevated cytokines, including interleukin 6 (IL-6), tumor necrosis factor alpha (TNF-α), interleukin 1 beta (IL-1β), interleukin 12 (IL-12) and transforming growth factor beta (TGF-β); group 2—unchanged cytokines, including interleukin 2 (IL-2), interleukin 4 (IL-4) and interleukin 17 (IL-17); group 3—elevated or unchanged cytokines, including interleukin 8 (IL-8) and interferon gamma (IFN-γ); group 4—interleukin 10 (IL-10) with increased, decreased and unchanged serum levels. However, this grouping is not unambiguous and includes mainly pro-inflammatory cytokines. In addition, the authors did not provide an analysis of the relationship between the levels of pro-inflammatory and anti-inflammatory cytokines in patients with TRS.

Higher serum levels of pro-inflammatory cytokines are characteristic of both patients with the first episode of Sch and patients with relapse and TRS, compared with the control group [80]. IL-1β, IL-6 and TGF-β were elevated at the first psychotic episode, and Sch flare normalized after AP treatment. Conversely, the levels of IL-12, IFN-γ, TNF-α and soluble interleukin 2 receptor (sIL-2R) remained elevated during exacerbations and during AP therapy [81]. A study of interleukins in the cerebrospinal fluid (CSF) showed that the levels of interleukin 6 (IL-6) and IL-8 were increased in Sch, but not significantly increased in affective disorders [82]. A meta-analysis of CSF cytokines showed higher levels of pro-inflammatory cytokines and lower levels of anti-inflammatory cytokines in patients with Sch and TRS [83].

It is known that dopaminergic dysfunction is a significant feature in the pathophysiology of TRS [84]. Interactions between cytokines and neurotransmitters in certain areas of the brain, and also during brain development, are important in the pathophysiology of TRS. Apparently, the pro-inflammatory cytokine IL-1β, which induces the transformation of rat mesencephalic progenitor cells into a dopaminergic phenotype [85], and IL-6, which reduces the survival of serotonergic neurons in the fetal brain, seem to play an important role in influencing neurotransmitter systems in TRS. [86]. Studies have found abnormalities in the cytokine system in patients with TRS [87,88]. There is evidence that the levels of IL-2 and IL-6 were elevated in patients with TRS, which is probably associated with the activation of the inflammatory response system (IRS). Moreover, serum IL-2 or IL-6 and cortisol are positively correlated with Sch, supporting the hypothesis that hypercortisolemia may also be caused by pro-inflammatory cytokines [89,90].

So, a summary of the results of studies on the role of neuroinflammation in the development of TRS is presented in Table 2.

## 3. Cytokines Alteration in Treatment-Resistant Schizophrenia 

Cytokines, which comprise a family of proteins—interleukins (IL), lymphokines, monokines, interferons and chemokines—are important components of the immune system (Table 3). 

Cytokines act in concert with specific cytokine inhibitors and soluble cytokine receptors to regulate the human immune response [91]. Their physiologic role in inflammation and pathologic role in systemic inflammatory states are now well recognized. An imbalance in cytokine production or cytokine receptor expression and/or dysregulation of a cytokine process contributes to various pathological disorders, including Sch [92]. Cytokines are classified as pro-inflammatory and anti-inflammatory. The time-dependent pro- and anti-inflammatory imbalance determines the outcome of an inflammatory response in development of TRS [93]. It should be clarified that the division of cytokines into pro- and anti-inflammatory is very conditional, because depending on the conditions, the cytokine can behave as a pro- or anti-inflammatory cytokine (for example, IL-6) [94]. Indeed, the number of cytokines, the nature of the activating signal, the nature of the target cell, the nature of the cytokines produced, the timing, the sequence of action of cytokines and even the experimental model are parameters that strongly affect the properties of cytokines [95].

### 3.1. Pro-Inflammatory Cytokines

Pro-inflammatory cytokines play a central role in neuroinflammatory disorders of infectious or noninfectious origin. Pro-inflammatory cytokines are produced predominantly by activated macrophages and are involved in the upregulation of inflammatory reactions [96]. These cytokines serve to contain and resolve the inflammatory foci through activation of local and systemic inflammatory responses. Pro-inflammatory cytokines may directly modulate neuronal activity in various classes of neurons in CNS, including dopaminergic neurons [97]. The major pro-inflammatory cytokines that are responsible for early responses are interleukin 1 alpha (IL1-α), interleukin 1 β (IL1-β), interleukin 6 (IL-6) and tumor necrosis factor alpha (TNF-α). Other pro-inflammatory mediators include members of the interleukin 20 (IL-20) family, interleukin 33 (IL-33), leukemia inhibitory factor (LIF), interferon gamma (IFN-γ), oncostatin M (OSM), ciliary neurotrophic factor (CNTF), transforming growth factor beta (TGF-β), granulocytic-macrophage colony-stimulating factor (GM-CSF), interleukin 11 (IL-11), interleukin 12 (IL-12), interleukin 17 (IL-17), interleukin 18 (IL-18), interleukin 18 (IL-8) and a variety of other chemokines that chemoattract inflammatory cells. These cytokines either act as endogenous pyrogens (IL-1, IL-6, TNF-α), upregulate the synthesis of secondary mediators and pro-inflammatory cytokines by both macrophages and mesenchymal cells, stimulate the production of acute phase proteins or attract inflammatory cells [98]. IL-1β, TNFα, IFN-γ, IL-12 and interleukin 11 (IL-18) are well characterized as pro-inflammatory cytokines. 

#### 3.1.1. Interleukin 1 β

IL-1β is produced by myeloid blood cells, pathogenic lymphocytes, resident microglia and CNS astrocytes in autoimmune diseases, neurodegeneration and metabolic diseases. It is a key pro-inflammatory cytokine that is involved in the regulation of the innate immune response [99]. IL-1β is a pleiotropic cytokine that can activate microglia and astrocytes and lead to subsequent synthesis of other pro-inflammatory cytokines and chemotactic mediators in the CNS [100]. IL-1β leads to aberrant release and accumulation of glutamate, which subsequently leads to neuronal death in most neurodegenerative diseases [101]. In a cross-sectional study by Enache et al. [102], investigating the association of plasma cytokine levels with TRS, no association of IL-1β with TRS was found. However, other studies have conflicting results [103].

#### 3.1.2. Tumor Necrosis Factor Alpha

TNF-α regulates several processes, including sleep, learning and memory, synaptic plasticity and astrocytic-induced synaptic reinforcement in the healthy CNS [103]. The biological functions of TNF-α are mediated through its two main receptors: tumor necrosis factor receptor 1 (TNFR1 or p55) and tumor necrosis factor receptor 2 (TNFR2 or p75). TNFR1 activation initiates inflammatory, apoptotic and degenerative cascades, while TNF-α signaling through TNFR2 is anti-inflammatory and cytoprotective, resulting in induction of proliferation, differentiation, angiogenesis and tissue repair [104]. TNF-α is also an important pro-inflammatory cytokine produced by both neurons and glial cells. Genetic association studies have provided evidence of Sch-associated gene variations in the innate and adaptive immune systems [105]. In a recent genetic study, which examined the relationship between TNF-α polymorphism–238 G/A and response to APs treatment, it was shown that while TNF-α polymorphisms–238 G/A and -308 G /A were not associated with Sch, TNF-α–238 G/A polymorphism may be associated with treatment resistance and suicide attempts in patients with Sch in the Turkish population [106]. Another study on the prognosis of TRS using immune-inflammatory biomarkers reported that TRS is associated with a specific cytokine-chemokine profile, i.e., elevated levels of C-C motif chemokine ligand 11 (CCL11), macrophage inflammatory protein-1 alpha (MIP-1α), soluble tumor necrosis factor receptors 1 (sTNF-R1) and soluble tumor necrosis factor receptors 2 (sTNF-R2), as well as decreased levels of interferon gamma induced protein 10 (IP-10), TNF-α, IL-2 and IL-4 [107]. Data from a 2021 crossover study showed, however, that both TRS and ultra-treatment-resistant Sch (UTRS) patients tended to increase TNFα expression, which, however, did not reach statistical significance [108]. 

#### 3.1.3. Interferon Gamma

IFN-γ is a soluble cytokine that is predominantly released from T helper type 1 (Th1), cytotoxic T lymphocytes and natural killer cells. IFN-γ serves to prime microglia, which is associated with various cellular adaptations, including changes in morphology, upregulation of receptors and increased levels of pro-inflammatory cytokines [38]. Data regarding the level of IFN-γ and TRS remain contradictory. For example, in a study by Upthegrove et al. [109], evidence has been obtained that elevated IFN-γ levels are associated with TRS. However, another study reported that IFN-γ was not associated with response to APs therapy [110].

#### 3.1.4. Interleukin 12

IL-12 secreted mainly by macrophages and dendritic cells in response to components of the bacterial cell wall. IL-12 stimulates proliferation, and also activates and increases the cytotoxicity of natural killer cells (NK cells) and T cells, promoting the differentiation of the latter into Th1. It is also known to induce the secretion of IFN-γ and TNF-α and has a synergistic effect with Interleukin 18 (IL-18) [92]. When examining the plasma level of IL-12, it was found to be elevated in TRS and UTRS [109].

#### 3.1.5. Interleukin 18

Research on the presence of IL-18 in the CNS began shortly after its discovery as a stimulator of inf-γ production in the immune system. IL-18 has been investigated for its similarity to IL-1β as a possible mediator of disease behavior and local inflammatory responses associated with neuronal injury. IL-18 promotes loss of appetite, sleep, and inhibition of long-term potentiation (LTP), and is also produced by and active in microglial cells and possibly contributes to neurodegeneration. IL-18 represents a link between the immune and nervous systems, since IL-18 and its receptors in the CNS mediate neuroinflammation by modulating homeostasis and behavior [111]. There is some evidence that IL-18 levels are elevated in patients with Sch but do not appear to be the cause of the disorder itself [112], although it is likely that elevated serum IL-18 levels may be a biomarker for TRS and UTRS.

#### 3.1.6. Interleukin 8

IL-8 is secreted predominantly in response to an antigen by macrophages, T-lymphocytes, neutrophils, and other cells; IL-8 is also the most potent human chemokine [113]. IL-8, being a pro-inflammatory cytokine, enhances the migration of neutrophils, T-lymphocytes and monocytes, whose enzymes produce free oxygen radicals and, thus, increase oxidative stress, which can lead to neuronal death [114]. Studies have shown that IL-8 significantly predicted non-response to APs therapy and positively correlated with negative Sch symptoms [102] and can be considered as a potential TRS biomarker.

#### 3.1.7. Interleukin 17 

IL-17 is secreted by helper lymphocytes 17 (Th17) and stimulates macrophages and microglia to secrete pro-inflammatory cytokines [115]. According to some data, no effect of APs on peripheral levels of IL-17 has been demonstrated [116]. However, it has been reported that activation of the IL-17 pathways may be present from the onset of Sch and appears to increase with disease progression up to the development of TRS and UTRS. The IL-23/IL-17 pathway is being considered as a therapeutic target for patients with TRS, especially since many anti-inflammatory drugs have been proposed as adjuncts to treat Sch symptoms, such as N-acetylcysteine, which appears to reduce they produce IL-17 [38].

Summary, the role of pro-inflammatory cytokines in TRS is presented in the Table 4.

### 3.2. Anti-Inflammatory Cytokines

The anti-inflammatory (immunosuppressive) cytokines are a series of immunoregulatory molecules that control the pro-inflammatory cytokine response. The anti-inflammatory cytokines act in concert with specific pro-inflammatory cytokine inhibitors and soluble cytokine receptors to regulate the human immune response. Major anti-inflammatory cytokines include interleukin 1 receptor antagonist (IL-1Ra), IL-4, IL-6, IL-10, IL-11, interleukin 13 (IL-13) and TGF-β. Specific cytokine receptors for IL-1, TNFα and IL-18 also function as pro-inflammatory cytokine inhibitors: IL-1Ra as an interleukin 1 alpha (IL-1α) and IL-1β antagonist; Interleukin-18-binding protein (IL-18BP) as an IL-18 antagonist [109]. Several newly found cytokines, such as IL-33, interleukin 35 (IL-35) and interleukin 37 (IL-37), also participate in regulating the function of neurons and neuroglia. Anti-inflammatory cytokines, in particular IL-10, inhibit pro-inflammatory cytokine synthesis and adhesion molecule expression, while increasing the levels of specific cytokine inhibitors [117]. IL-1Ra, IL-4, IL-6 and interleukin 10 (IL-10) are well characterized as anti-inflammatory cytokines. 

#### 3.2.1. Interleukin 4

IL-4 is produced by activated Th lymphocytes, mainly Th2 helper lymphocytes, natural killer T cells (NK cells), mast cells and basophils. Its role is to promote the differentiation of Th into Th2 lymphocytes, as well as to increase their cytotoxicity. IL-4 affects macrophages and microglial cells and may be neuroprotective by reducing their ability to induce oxidative stress. In addition, IL-4 also plays a role in cognitive processes [118]. A study of serum IL-4 levels in patients with TRS did not reveal a significant difference in IL-4 levels between the three groups—patients with TRS, patients without TRS and a healthy control group [119]. In another study, Şükrü et al. [120] also found no significant differences between patients with TRS and the control group in terms of serum IL-4 levels.

#### 3.2.2. Interleukin 6

IL-6 is a multifunctional pro-inflammatory cytokine that is secreted primarily through monocytes and macrophages. They play a key role in processes related to immunity and neuroinflammation. IL-6 regulates the transmission of neuronal excitability, metabolism and sensitivity of CNS neurotransmitters to them. High concentrations of IL-6 at baseline have been associated with the development of TRS and prolonged hospital stay [121]. Mongan et al. [122] showed that serum IL-6 was significantly higher in TRS patients than in normal volunteers, while Sch patients without TRS showed intermediate values. In addition, elevated levels of IL-6 were found in not only patients with TRS, but also in patients with UTRS, confirming the significant predictive role of IL-6 in TRS. IL-6 is one of the main effector cytokines of Th17 cells. However, it is also possible that the high levels of IL-6 observed in the UTRS subgroup may indicate a bias towards the Th2 pathway associated with the chronicity of the schizophrenic process [123]. TRS is also accompanied by signs of IRS and compensatory immunoregulatory system (CIRS), including activation of M1 cells (especially IL-6 and TNFα trans-signaling) [96]. Clear immune abnormalities are seen in TRS patients, and IL-6 may be an important marker of TRS [124].

#### 3.2.3. Interleukin 10 

IL-10 and receptors to IL-10 (IL-10R) are synthesized in the CNS, including by microglia and astrocytes; they can be considered an important modulator of neuroinflammation [125]. After IL-10 binds to its receptor, this cytokine initiates its cellular effects through canonical Janus kinase (JAK)/signal transducer and transcriptional activator (STAT), which includes JAK1 and STAT3, which subsequently induces the expression of genes associated with immunosuppression [126]. Taken together, the evidence suggests that IL-10 plays a critical role in limiting inflammation in the CNS, similar to that seen in peripheral sites, by altering the ability of resident glia and infiltrating leukocytes to respond to activating stimuli and decreasing the production of inflammatory mediators by these cells [127]. Patients with TRS show high levels of IL-10. The upregulation of this potent anti-inflammatory cytokines may reflect the induction of contractive homeostatic processes [38]. A meta-analysis by Marcinowicz et al. [116] demonstrated a decrease in serum IL-10 levels in patients with a first psychotic episode after APs.

A summary of the role of anti-inflammatory cytokines in TRS is presented in the Table 5.

## 4. Correction of Cytokine Status Imbalance as a Promising Therapeutic Strategy for Treatment-Resistant Schizophrenia

Current treatment of Sch considers the use of APs as the first line of therapy [128,129]. This is often followed by the use of non-steroidal anti-inflammatory drugs (NSAIDs) in addition to nutrients (vitamins, minerals, plant and animal products) that affect inflammation and the immune system [130]. The effect of APs and NSAIDs on the level of cytokines has also been shown with a decrease in the expression levels of pro-inflammatory cytokines, such as IL-18, IL-1β, IL-6 and IL-8 [131,132].

The heterogeneity of the phenotypes underlying Sch and the high likelihood of adverse drug reactions (ADRs) with APs support the need to explore new treatment strategies for TRS, most of which are under study. Therapy for TRS with monoclonal antibodies, intravenous immunoglobulins (IVIG), NSAIDs, corticosteroids, tetracycline antibiotics, antioxidants and statins have been described as possible strategies for disease-modifying therapy [130]. Prospects also have a correction of the cytokine imbalance in patients with TRS. Thus, this issue remains open and is actively studied in connection with the relevance of the task (Table 6).

## 5. Discussion

The delicate balance between pro-inflammatory and anti-inflammatory cytokines determines the net effect of a neuroinflammatory response in patients with TRS. Perturbations in this equilibrium can drive the patient defense immune response towards chronic neuroinflammation (pro-inflammatory) or towards healing (anti-inflammatory). Thus, a cytokine imbalance may be beneficial to the patient with Sch by initiating the neuroinflammatory response. However, overproduction or underproduction of pro-inflammatory or anti-inflammatory endogenous mediators (cytokines) may actually be deleterious to the patient with “non-dopamine’’ Sch (Figure 4). 

In addition, chronic neuroinflammation, supported by an imbalance between pro-inflammatory and anti-inflammatory cytokines, and persistent dopaminergic neurotransmission disorder can be considered as an overlap syndrome in patients with “dopamine” Sch.

A genetic predisposition that determines the balance of pro-inflammatory and anti-inflammatory cytokines and, hence, susceptibility to TRS is very important in the patients with Sch. Various single nucleotide variants (SNVs) have been identified within pro-inflammatory and anti-inflammatory cytokine and cytokine receptor genes that alter their expression. These SNVs of cytokine and cytokine receptor genes may determine the imbalance of pro-inflammatory and anti-inflammatory cytokines in the neuroinflammatory response in patients with “non dopamine” and “dopamine” Sch.

To date, therapeutic strategies targeting pro-inflammatory cytokines may be effective in treating TRS. Pro-inflammatory cytokines are known to be crucial for initiating a neuroinflammatory response. However, their level in the CNS may have reached its absolute or relative peak before the clinical signs of TRS became apparent.

In addition, therapy that blocks pro-inflammatory cytokines, paradoxically, may lead to increased inflammation [151]. Various inflammation paradoxes have been reported, including new inflammations ocurring when: (1) particular cytokine and inflammatory regulators encoding genes are mutated [152]; (2) patients experience somatic mutations [153] and inflammation [154]; (3) pro-inflammatory cytokines are weakened due to SNVs of the genes encoding them [155]; (4) pro-inflammatory cytokine blockage therapies are used; etc. [144]. Pro-inflammatory cytokines and regulators are interconnected through evolution; single cytokine blockade therapies may result in significant upregulation of a long list of genes and signaling pathways, presumably the “second wave of inflammation” [156]. For example, the second wave of inflammation may be the main mechanism of ADRs observed in patients receiving Mab therapy that blocks pro-inflammatory cytokines [151].

Nevertheless, our narrative review provides a new insight into the role of imbalance between pro-inflammatory and anti-inflammatory cytokines in the pathogenesis of TRS and in new approaches to predicting and early diagnosis of the development of TR to APs, as well as new targets for future therapeutic interventions in “non-dopamine” Sch.

## 6. Conclusions

Our narrative review demonstrates that the problem of evaluating the contribution of pro-inflammatory and anti-inflammatory cytokines to maintaining or changing the cytokine balance can become a new key in unlocking the mystery of “non-dopamine” Sch and developing new therapeutic strategies for the treatment of TRS and psychosis in acute and chronic neuroinflammation. In addition, the inconsistency of the results of previous studies on the role of pro-inflammatory and anti-inflammatory cytokines indicates that the TRS biomarker, most likely, is not the serum level of one or several cytokines, but the cytokine balance. We have demonstrated a hypothesis that the cytokine imbalance is one of the most important TRS biomarkers. Partially, this hypothesis is supported by the variable response to immunomodulators in patients with TRS, which were prescribed without taking into account the cytokine balance of the relation between serum levels of the most important pro-inflammatory and anti-inflammatory cytokines for TRS.

## Figures and Tables

**Figure 1 ijms-23-11324-f001:**
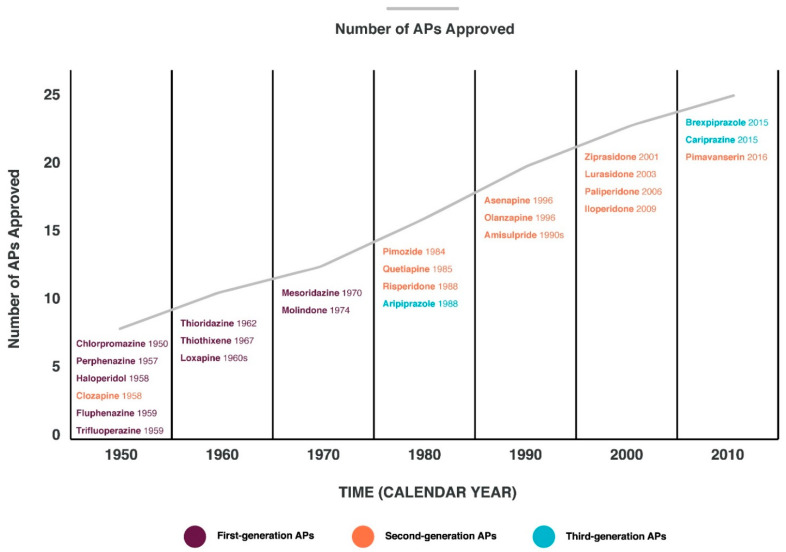
Timeline of antipsychotics (APs) approved by the Food and Drug Administration (FDA) [19].

**Figure 2 ijms-23-11324-f002:**
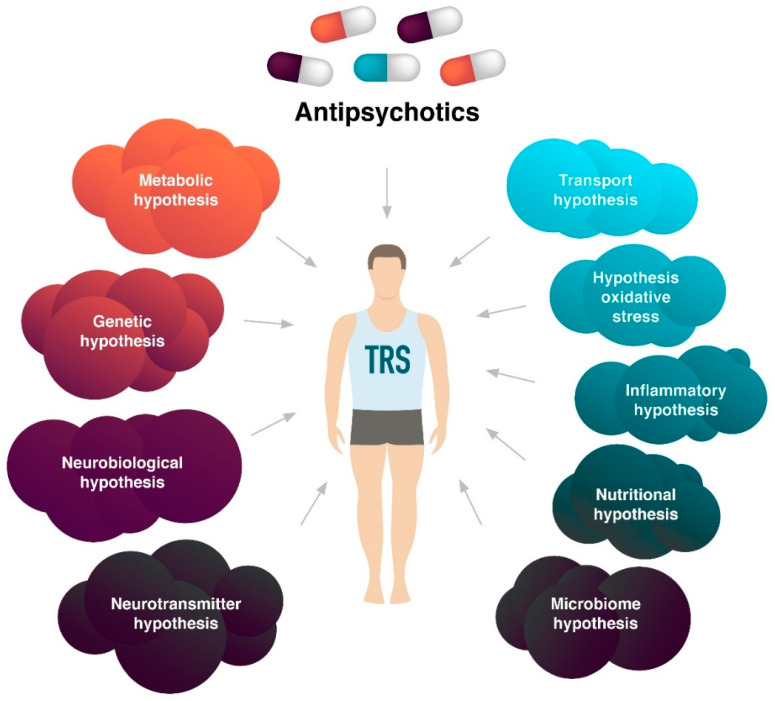
Hypotheses of treatment-resistant schizophrenia (TRS).

**Figure 3 ijms-23-11324-f003:**
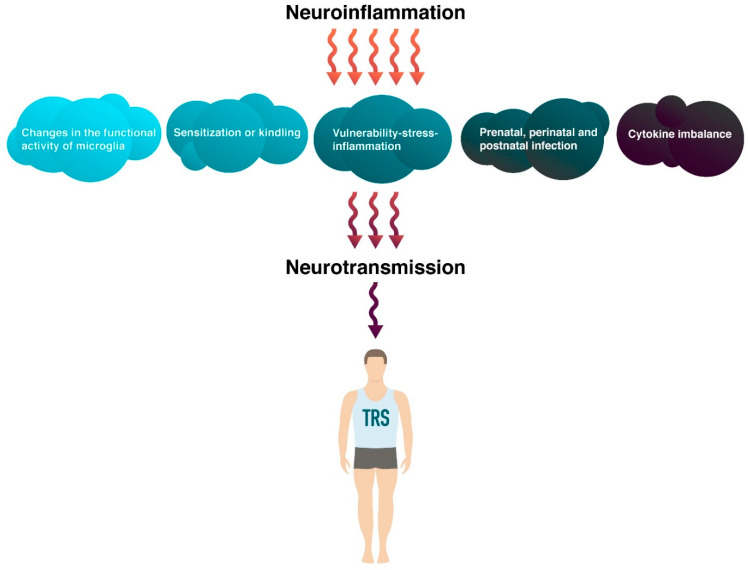
Mechanisms of neuroinflammation that are associated with the treatment-resistant schizophrenia (TRS).

**Figure 4 ijms-23-11324-f004:**
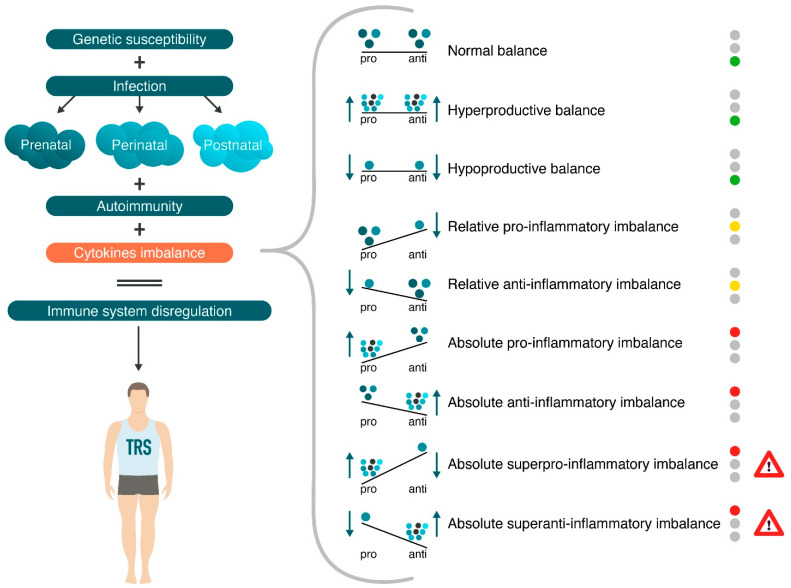
Potential role of normal and abnormal cytokine levels in the cytokine imbalance as a biomarker of treatment-resistant schizophrenia (TRS). Note: green—low risk; yellow—middle risk; red—high risk; red + attention sign—very high risk.

**Table 1 ijms-23-11324-t001:** Hypotheses for the development of treatment-resistant schizophrenia.

Hypothesis	Mechanism	References
Genetic	Genetic predisposition to low affinity of targets (dopaminergic receptors) to APs of the first and new generations.	[22,23]
Neurodevelopmental	Congenital minor anomalies of brain development (microdysgenesis) in brain regions critical for Sch development.	[24,25]
Neurotransmitter	Violation of the synthesis, release, or breakdown of dopamine and other neurotransmitters (serotonin, melatonin, etc.).	[19,26,27,28]
Metabolic	Primary (genetically determined) and secondary metabolic disorders of APs of the first and new generation in the liver.	[29,30,31]
Transport	Primary (genetically determined) and secondary impairment of expression and/or functional activity of APs transporter proteins of the first and new generations across the blood–brain barrier.	[32,33,34]
Oxidative stress	Violation of the prooxidant-antioxidant balance in favor of the former, which leads to oxidative damage to cellular lipids, proteins, enzymes, carbohydrates and DNA, which contributes to a worsening of the course and an unfavorable outcome of Sch.	[35,36,37]
Inflammatory	Primary (genetically determined) and secondary violation of the cytokine status (absolute or relative hyperproduction of pro-inflammatory cytokines).	[38,39,40]
Microbiome	Microbiota through the gut–brain axis is associated with the development and severity of Sch, intestinal microbiota is associated with the response to APs.	[41,42,43]
Nutritional	Deficiency or excess of nutrients (vitamins, minerals, amino acids) necessary for the functioning of the dopaminergic system of the brain.	[44,45,46]

Note: APs—antipsychotics; DNA—deoxyribonucleic acid; Sch—schizophrenia.

**Table 2 ijms-23-11324-t002:** Studies of the role of neuroinflammation in the development of treatment-resistant schizophrenia.

Author, Year	Mechanism	Pathogenesis	Reference
Meehan et al., 2017	Prenatal, perinatal and postnatal infection	Immune activation. Violation of neurogenesis processes, including dopaminergic and glutamatergic neurotransmission.	[75]
Frank et al., 2018	Sensitization or kindling	Stimulation of the immune response. Activation of cell proliferation, increased production and release of pro-inflammatory cytokines.	[57]
Momtazmanesh et al., 2019	Cytokine imbalance	Increased serum levels of pro-inflammatory cytokines IL-1β, IL-6 and TGF-β.	[79]
Wang et al., 2020	Cytokine imbalance	Interactions between cytokines and neurotransmitters in certain areas of the brain, as well as during brain development. Induction of IL-1β conversion of mesencephalic progenitor cells into a dopaminergic phenotype. Reduced survival of serotonergic neurons through IL-6.	[83]
Kumar et al., 2020	Sensitization or kindling	Stimulation of the glutamatergic system, ionotropic and metabotropic glutamate receptors that excite amino acid transporters. Increased levels of glutamate in the anterior cingulate cortex.	[61]
Woodburn et al., 2021	Changes in the functional activity of microglia	Priming of microglia causes an exaggerated immune response. Proliferation and increased production of pro-inflammatory cytokines.	[52]
Müller et al., 2021	Prenatal, perinatal and postnatal infection	Increased levels of CRP and pro-inflammatory cytokines in childhood.	[76]
Dziurkowska et al., 2021	Cytokine imbalance	Increased plasma levels of IL-2 and IL-6, activation of IRS. Positive correlation of IL-2, IL-6 and cortisol, hypercortisolemia.	[89]
Woodburn et al., 2021	Sensitization or kindling	Pro-inflammatory immune response in the CNS. Activation and proliferation of microglia. Mediated neurotransmitter disorders.	[52]
Rovira et al., 2022	Prenatal, perinatal and postnatal infection	Violation of the structure, exposure to inflammatory factors, neurochemical changes. Increased dopamine levels, impaired COMT activity.	[66]

Note: CNS—central nervous system; COMT—catechol-O-methyl transferase; CRP—C-reactive protein; IL-1β—interleukin 1 β; IL-2—interleukin 2; IL-6—interleukin 6; IRS—inflammatory response system; TGF-β—transforming growth factor beta.

**Table 3 ijms-23-11324-t003:** Pro-inflammatory and anti-inflammatory cytokines.

Pro-Inflammatory Cytokines	Anti-Inflammatory Cytokines
Ciliary neurotrophic factor (CNTF) Granulocytic-macrophage colony-stimulating factor (GM-CSF) Interferon gamma (IFN-γ) Interleukin 20 (IL-20) Interleukin 1 alpha (IL1-α) Interleukin 1 β (IL1-β) Interleukin 11 (IL-11) Interleukin 12 (IL-12) Interleukin 17 (IL-17) Interleukin 18 (IL-18) Interleukin 18 (IL-8) Interleukin 33 (IL-33) Interleukin 6 (IL-6) Leukemia inhibitory factor (LIF) Oncostatin M (OSM) Transforming growth factor beta (TGF-β) Tumor necrosis factor alpha (TNF-α)	Interleukin 1 receptor antagonist (IL-1Ra) Interleukin 10 (IL-10) Interleukin 11 (IL-11) Interleukin 13 (IL-13) Interleukin 4 (IL-4) Interleukin 6 (IL-6) Interleukin-18-binding protein (IL-18BP) Transforming growth factor beta (TGF-β)

**Table 4 ijms-23-11324-t004:** Role of pro-inflammatory cytokines in treatment-resistant schizophrenia.

Cytokine	Gene: OMIM	Role in Neuroinflammation	Role in TRS	References
IL-1β	*IL1B*: 147720	Stimulation of the synthesis of other pro-inflammatory and chemotactic mediators in the CNS. Stimulation of aberrant release and accumulation of glutamate, which subsequently leads to neuronal death in most neurodegenerative diseases.	+/− or +	[99,100,101,102]
TNF-α	*TNF**A*: 191160	Regulation of several processes including sleep, learning and memory, synaptic plasticity and astrocytic-induced synaptic strengthening. Initiation of inflammatory, apoptotic and neurodegenerative cascades, while TNF-α signaling via TNFR2 is anti-inflammatory and cytoprotective, resulting in induction of proliferation, differentiation, angiogenesis and tissue repair.	+++	[38,104,105,106,107,108]
IFN-γ	*IFNG:* 147570	Priming of microglia, which is associated with various cellular adaptations, including changes in morphology, upregulation of receptors and an increase in pro-inflammatory cytokines.	+/−	[102,109,110]
IL-12A	*IL12A*: 161560	Stimulation of proliferation. Activation and increase in the cytotoxicity of NK cells and T cells. Stimulation of differentiation in Th1. Induction of IFN-γ and TNF-α secretion, synergism with pro-inflammatory cytokines with IL-18.	+++	[38,92]
IL-18	*IL18*: 600953	Potentiation of the development of the relationship between the immune and nervous systems, since IL-18 and its receptors in the CNS mediate neuroinflammation of the brain, modulating homeostasis and behavior.	++	[111,112]
IL-8	*CXCL8:* 146930	Increased migration of neutrophils, T cells and monocytes, whose enzymes produce free oxygen radicals Indirect increase in oxidative stress, which can lead to neuronal death.	+++	[102,113,114]
IL-17	*IL17A*: 603149	Stimulation of macrophages and microglia to secrete pro-inflammatory cytokines in the CNS.	+++	[38,109,115]

Note: (+/−)—questionable prognostic role in the development of TRS; (+)—mild prognostic role in the development of TRS; (++)—moderate prognostic role in the development of TRS; (+++)—significant prognostic role in the development of TRS; CNS—central nervous system; IFN-γ—interferon gamma; IL-12—interleukin 12; IL-17—interleukin 17; IL-18—interleukin 18; IL-1β—interleukin 1 β; IL-8—interleukin 8; NK cells—natural killer cells; T cells—T-lymphocytes; Th1—type 1 helper T cells; TNFR2—tumor necrosis factor receptor 2; TNF-α—tumor necrosis factor alpha.

**Table 5 ijms-23-11324-t005:** Role of anti-inflammatory cytokines in treatment-resistant schizophrenia.

Cytokine	Gene: OMIM	Role in Neuroinflammation	Role in TRS	References
IL-4	*IL4: 147780*	Initiation of T-helper differentiation into T-helper 2 lymphocytes. Increased Th2 cytotoxicity. Modulation of the function of macrophages and microglial cells. Decreased cytotoxicity.	+/−	[118,119]
IL-6	*IL6:* 147620	A key role in the processes associated with immunity and neuroinflammation. Modulation of the sensitivity of neurons to neurotransmitters.	+++	[120,121,122,123,124]
IL-10	*IL10:* 124092	Initiation of cellular effects through canonical JAK/ STAT, which includes JAK1 and STAT3. Induction of expression of genes associated with immunosuppression.	+++	[38,116,125,126,127]

Note: +/−—mild prognostic role in the development of TRS; +++—significant prognostic role in the development of TRS; IL-10—interleukin 10; IL-10R1—inter-leukin-10 receptor 1; IL-10R2—interleukin-10 receptor 2; IL-4—interleukin 4; IL-6—interleukin 6; JAK—Janus kinase; JAK1—Janus kinase 1; STAT—signal transducer and activator of transcription; STAT3—signal transducer and activator of transcription 3.

**Table 6 ijms-23-11324-t006:** Perspective strategies for anti-inflammatory therapy in treatment-resistant schizophrenia.

Drug	Mechanism	Results	References
*Non-steroidal anti-inflammatory drugs*			
Celecoxib	Selective inhibition of COX-2.	Significant reduction in PANSS positive TRS symptom scores and overall PANSS score, but no significant change in negative TRS symptoms. Improvement in conceptual disorganization and abstract thinking by PANSS in patients with TRS.	[133,134]
Acetylsalicylic acid	Inhibition of COX-1 and c COX-2.	Improvement in PANSS symptoms.	[135,136]
*Statins*			
Simvastatin	Inhibition of HMG-CoA reductase, anti-inflammatory effect, reduction of pro-inflammatory cytokines (IL-1β, IL-6, TNF-α) and CRP.	Decrease in negative symptom scores on the PANSS scale in patients with TRS, decrease in the total score on the PANSS scale.	[137]
Pravastatin	Inhibition of HMG-CoA reductase, anti-inflammatory effect, reduction of pro-inflammatory cytokines (IL-1β, IL-6, TNF-α) and CRP.	Marked decrease in scores positive symptoms on the PANSS scale.	[138]
*Corticosteroids*			
Cortisone	Influence on carbohydrate and electrolyte metabolism, anti-inflammatory (inhibition of phospholipase A2), desensitizing and anti-allergic, immunosuppressive effects.	Most patients with Sch did not show significant changes in Sch symptoms.	[139]
Prednisolone	Suppression of the function of leukocytes and tissue macrophages. Limitation of migration of leukocytes to the area of inflammation, impairment of the ability of macrophages to phagocytosis, as well as to the formation of IL-1, inhibition of the activity of phospholipase A2, suppression of the release of COX-1 and COX-2, etc.	There was no significant difference in improvement in the severity of Sch symptoms with the placebo group in patients with Sch.	[140]
*Monoclonal antibody*			
Tocilizumab	Selective binding and suppression of expression and functional activity of IL-6 receptors.	No significant change in scores for positive and negative TRS symptoms, but improvement in BACS cognition.	[141,142]
*Cytokines*			
- IFN-γ-1b	Activation of macrophages and induction of expression of the class II major histocompatibility complex molecule, inhibition of virus replication.	A pronounced decrease in the total PANSS score in patients with TRS.	[143]
*Intravenous immunoglobulins*			
- IgG	Increasing the content of antibodies in the blood to a physiological level, creating passive immunity.	A pronounced decrease in the total PANSS score in patients with antibody positive psychosis. Most patients gave a clinical response to therapy.	[144,145]
*Other groups of drugs*			
Mucolytics/antioxidants: - N-acetylcysteine	Precursor of the biological antioxidant glutathione, anti-inflammatory and antioxidant effect.	A decrease in scores on all three PANSS scales, an improvement on the CGI-S, CGI-I scales in patients with TRS. The reduction in negative symptom scores on the PANSS scale was more significant in patients with TRS.	[146,147]
Antibiotics: - Minocycline	Bacteriostatic action due to the suppression of protein synthesis by reversible binding to the 30S ribosomal subunit of sensitive microorganisms.	Decrease in scores on all three PANSS scales, improvement in BPRS scores, no changes in cognitive function in patients with TRS.	[148]
Polyunsaturated fatty acids: - Omega-3 fatty acids	Antioxidant, anti-inflammatory and neuroprotective effect.	Significant improvement on the three PANSS scales, as well as improvement in cognitive functions, was not revealed.	[149,150]

Note: BACS—Brief Assessment of Cognition in Schizophrenia; BPRS—Brief Psychiatric Rating Scale; CGI-I—Clinical Global Impression—Improvement; CGI-S—Clinical Global Impression—Severity; COX-1—cyclooxygenase-1; COX-2—cyclooxygenase-2; CRP—C-reactive protein; HMG-CoA—3-hydroxy-3-methyl-glutaryl-coenzyme A reductase; IFN-γ-1b—interferon-γ-1b; IgG—immunoglobulins G; PANSS—Positive and Negative Syndrome Scale.

## Data Availability

Not applicable.

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
