# Peer review of "Cytokine Imbalance as a Biomarker of Treatment-Resistant Schizophrenia"

_ijms, 2022, doi:10.3390/ijms231911324_

Round 1

Reviewer 1 Report

Treatment resistant schizophrenia is a very important and difficult issue, therefore the prediction of treatment response and possible markers of treatment resistance could be very helpful in the treatment plan. The neuroinflammation is a very up to date issue in psychiatry, especially in schizophrenia and depression, but still new information is needed to clarify the possible mechanisms underlying the pathophysiology of mental diseases There are only a few reviews analyzing the markers of treatment resistance, especially including the inflammatory markers In my opinion the conclusions are very informative and summarize the evidence presented in the paper. References are appropriate.Tables and figures are very informative and clear

Author Response

Dear Reviewer,

Thank you very much for your interest in our manuscript and high marks.

Sincerely,

Authors

Reviewer 2 Report

Manuscript Summary:

This review provides a detailed outline of current treatment-resistant schizophrenia diagnostic definitions and their limitations. The authors provide a breakdown of antipsychotic medication development over time and pinpoint the lack of progress towards overcoming therapeutic resistances to schizophrenia (TRS) specifically. The paper provides a strong collection of primary and secondary sources to establish the relationship between inflammation, inflammatory regulation, and schizophrenia. Background information is provided on the critical aspects of the immune system potentially related to schizophrenia – microglia, cytokines and chemokines are overall well-defined. The sections detailing the importance of stress-induced and prenatal, perinatal, and post-natal infections were especially informative and the authors provide insightful analysis of current paradigms in schizophrenia research. There is a substantial list of references provided that express the importance of pro-inflammatory and anti-inflammatory effectors in schizophrenia disease onset and severity. The authors discuss the potential benefits and drawbacks to the immunomodulatory approach dependent on the cause of targeted schizophrenia. Nevertheless, there are several issues with the current manuscript that require revision prior to publication..

Major Comments: 

1.     The title of the review should be narrowed to be specifically inidicate therapeutic resistance to schizophrenia. 

2.     The authors should briefly describe Sch – they provide an exhaustive description of how to classify TRS but never start with Sch itself. 

3.     Prior to describing TRS, the authors should generally describe how 1st to 3rd generation antipsychotics function with respect to dopamine. This will provide the context for why inflammatory processes are so important to consider for TRS.

4.     In general, there is a significant amount of duplication in the information between the text and the tables. This reviewer suggests reorganizing several tables. The authors could effectively communicate the comparisons of the different TRS definitions is a table instead of the text. Table 2 should be removed since it is redundant with section 2. Tables 4 and 5 are not specific to TRS (despite the titles) – this reviewer suggests changing the title of the tables to include the generic cytokine information. The corresponding text (subsections of 3.1 and 3.2) should be revised to focus specifically on the TRS-specific information for the prioritized cytokines. This would remove a great deal of redundancy. 

5.     Figure 5 should be removed as the figure legend does not provide adequate detail explaining the figure and the text does not really touch on this subject matter. Similar, more detail should be added to the Figure 4 legend because this is the crux of the authors indicated goal for this manuscript.

6.     The authors make a provocative statement that “they have proposed for the first time a new hypothesis that cytokine imbalance is one of the most important TRS biomarkers. It is unclear how the review supports this statement. To this reviewer it appears that the authors have highlighted that a single cytokine biomarker is not sufficient to account for prognosis of TRS. While that is correct, it is not clear how this is evidence of the authors strong statement. This reviewer suggests that the authors soften their claim.

7.     The background in Section 3 would be more helpful if provided before section 2.5.

Minor comments:

1.     Line 118. The text should refer to Figure 3, not Figure 2. 

2.     Figure 2 legend – unclear what Hypothesis integrating factors are

3.     It is unclear if section 2.2 is specific to TRS or general to Sch – this is a repeating issue in the subsections throughout the review

4.     Line 179. The authors should soften their statement to “…the offspring MAY BE at risk of developing Sch” instead of ARE.

5.     The groupings in section 2.5 are very ambiguous. Are the groupings based upon specific cytokines (e.g. Group 4 specific to IL-10), or are the groupings referring to the direction of change (e.g. elevated or unchanged).

6.     Line 310 – reference 102 (Enache et al) should be ref. 95 – this causes a mis-numbering of subsequent references.

7.     There are multiple grammatical errors throughout the manuscript.

Author Response

Dear Reviewer,

Thank you very much for your comments and recommendations for improving our manuscript.

Major Comments:

1) The title of the review has been modified.

2) Information about schizophrenia has been added.

3) Information about the effect of antipsychotics on dopaminergic receptors has been added.

4) Comment 4: the authors suggest leaving the tables, since they not only summarize the information by sections, but also allow to increase the visibility of this review in Internet resources. In addition, two other reviewers have no comments on the design and content of these tables.

Comment 5: Figure 5 has been removed.

Comment 6: The authors have softened their claims.

Comment 7: The authors suggest leaving background information about cytokines at the beginning of section 3, as this makes it easier for readers to understand the essence of the further analyzed studies.

Minor Comments:

1) Fig. 2 presents hypotheses, not mechanisms of the development of therapeutically resistant schizophrenia. Therefore, the authors placed the figures sequentially.

2) The legend of Fig. 2 has been changed.

3) The authors analyzed the problems of the development of schizophrenia, which, according to some studies, could also be a problem of the development of therapeutic resistance to antipsychotics in schizophrenia. Indeed, this boundary is very thin, and it needs further study.

4) Corrected.

5) We agree with the reviewer. In this case, we cited the grouping of cytokines proposed by Momtazmanesh et al. (Momtazmanesh, S.; Zare-Shahabadi, A.; Rezaei, N. Cytokine alterations in schizophrenia: An updated review. Frontiers in Psychiatry 2019, 10, 892.). We, like the reviewer, do not fully understand this grouping. We have added a comment to this grouping. Therefore, we further propose to consider cytokines from the position of their grouping into pro-inflammatory and anti-inflammatory. 

6) Corrected

7) Corrected

Thank you very much again.

Sincerely,

Authors

Reviewer 3 Report

This review describes the role of cytokine imbalance in the development and progression of treatment-resistant schizophrenia. Overall, the review is well organized and has a sufficient number of references. The topic is not novel. A review was recently (2021) published on the role of cytokines in schizophrenia development (PMID: 33746786). However, this manuscript focuses on the more specific area of treatment-resistant schizophrenia and provides additional details, as evident in Tables 2, 4, 6, and Fig 4. Because of this, the current review retains the factor of novelty. 

Overall, the reviewer suggests minor modifications. 

  1. The sentence on page 6, line 150, is not completed. Probably, it should be edited as "...an increase in the production and release of pro-inflammatory cytokines is observed."
  2. Figures 3 and 5 have the line underneath: "Note: TRS – treatment-resistant schizophrenia." It is not needed. 
  3. Figure 5 needs more details in the legend. 

Author Response

Dear Reviewer,

Thank you so much for your interest in the topic of therapeutically resistant schizophrenia and our manuscript.

We have modified the manuscript to take into account your comments:

Comment 1: the proposal has been supplemented.

Comment 2: The note has been deleted.

Comment 3: The figure was removed on the recommendation of the second reviewer.

Sincerely,

Authors

Round 2

Reviewer 2 Report

The authors have addressed this reviewer's major concerns.